# RLBenchNet: Benchmarking Neural Architectures with PPO Across Reinforcement Learning Tasks

## Abstract

Reinforcement learning (RL) has advanced significantly through the application of diverse neural network architectures. In this study, we systematically evaluate the performance of several architectures within RL tasks using a widely adopted policy gradient algorithm, Proximal Policy Optimization (PPO). The architectures considered include Long Short-Term Memory (LSTM), Multi-Layer Perceptron (MLP), Mamba/Mamba-2, Transformer-XL, Gated Transformer-XL, and Gated Recurrent Unit (GRU). Through comprehensive experiments spanning continuous control, discrete decision-making, and memory-based environments, we uncover architecture-specific strengths and limitations. Our results show that: (1) MLPs excel in fully observable continuous control tasks, offering an effective balance between performance and efficiency; (2) recurrent architectures such as LSTM and GRU provide robust performance in partially observable settings with moderate memory demands; (3) Mamba models achieve up to 4.5× higher throughput than LSTM and 3.9× higher than GRU, while maintaining comparable performance; and (4) only Transformer-XL, Gated Transformer-XL, and Mamba-2 succeed on the most memory-intensive tasks, with Mamba-2 requiring 8× less memory than Transformer-XL. These findings highlight the trade-offs among architectures and provide actionable insights for selecting appropriate models in PPO-based RL under different task characteristics and computational constraints. Code is available at: `https://anonymous.4open.science/r/BenchNetRL-A718`

## 1 Introduction

Reinforcement learning (RL) has emerged as a powerful framework for sequential decision-making, with neural networks playing a key role in enabling agents to learn complex policies (Gu et al., 2023; 2025; Silver et al., 2016). Despite their importance, the influence of neural network architecture on RL performance across diverse environments remains relatively underexplored in the literature. In this work, we aim to bridge this gap by systematically evaluating the impact of various neural network architectures on the performance of RL agents.

To ground our investigation, we choose Proximal Policy Optimization (PPO) (Schulman et al., 2017) as the primary algorithm for our study, as it is one of the most widely adopted RL methods, known for its simplicity and strong empirical performance. Our experiments span a diverse set of benchmark tasks, offering comprehensive insights into how architectural choices affect learning efficiency and policy quality in RL. These tasks include environments requiring memory, such as Partially Observable Markov Decision Processes (POMDPs), and environments focused on continuous control and discrete decision-making. By analyzing the strengths and weaknesses of architectures such as LSTM (Hochreiter & Schmidhuber, 1997), GRU (Chung et al., 2014), Transformer-XL (Dai et al., 2019), Gated Transformer-XL (GTrXL) (Parisotto et al., 2019), Mamba (Gu & Dao, 2024), Mamba-2 (Dao & Gu, 2024) and MLP, we aim to provide actionable insights into the design of RL systems.

Previous works have explored implementation details of PPO (Huang et al., 2022a), that methods could be used to improve the agent performance in various environments (Andrychowicz et al., 2020), and studies like (Pleines et al., 2024) have demonstrated the efficacy of Transformer-XL in episodic memory tasks. However, these studies often overlook comparisons with simpler architectures, such

as LSTM, and emerging architectures, like Mamba. Furthermore, existing benchmarks, such as those conducted in Memory Gym (Pleines et al., 2024) and MiniGrid (Chevalier-Boisvert et al., 2023), highlight the need for memory in certain environments but lack comprehensive comparisons across a broader range of architectures and tasks.

Our contributions are threefold:

- **Architecture Benchmarking in RL:** We benchmark PPO implementations using a variety of neural network architectures, including traditional models (MLP, LSTM, GRU) and advanced architectures (Transformer-XL, GTrXL, Mamba, Mamba-2). We also provide insights into selecting appropriate neural networks for different settings.

- **Evaluation Across Diverse Tasks:** We evaluate these neural network architectures across memory-intensive environments such as MiniGrid; partially observable classical control tasks, including LunarLander, Acrobot, and CartPole; and both continuous and discrete control domains, such as MuJoCo (Todorov et al., 2012) and Atari (Bellemare et al., 2013). We further provide a comprehensive analysis of the experimental results.

- **Trade-off and Guideline Analysis:** We analyze the trade-offs between memory requirements, computational efficiency, and task performance, offering practical guidelines for selecting architectures based on task characteristics.

## 2 RELATED WORK

### 2.1 MEMORY MODELS FOR REINFORCEMENT LEARNING

Memory modeling in RL has evolved through three main architectural paradigms: *recurrent networks*, *transformer-based models*, and *state-space models*. Each of these architectures offers distinct advantages depending on the task characteristics, particularly in environments with varying levels of partial observability and memory requirements.

**Recurrent networks**, including LSTM and GRU (Hochreiter & Schmidhuber, 1997; Chung et al., 2014), have been the traditional choice for handling partial observability in RL. These architectures maintain an internal memory state that is updated at each timestep, enabling them to integrate information over time. Studies like MemoryGym (Pleines et al., 2024) have shown that well-tuned GRU models can outperform even advanced architectures like Transformer-XL in indefinite-horizon tasks (Pleines et al., 2023). However, their effectiveness is limited in long-horizon scenarios due to vanishing gradient issues and fixed memory capacity (Lu et al., 2024).

**Transformer-based architectures**, such as Transformer-XL and GTrXL (Dai et al., 2019; Parisotto et al., 2019), introduce self-attention mechanisms that excel at modeling long-term dependencies. These models have demonstrated state-of-the-art performance in partially observable environments (Ni et al., 2023), but their quadratic complexity with sequence length can become a computational bottleneck. Hybrid models, like the Recurrent Memory Transformer (RMT) (Cherepanov et al., 2024) and Recurrent Action Transformer with Memory (RATE) (Bulatov et al., 2024), extend transformer capabilities by integrating external memory mechanisms.

**State-space models**, particularly Mamba and Mamba-2 (Gu & Dao, 2024; Dao & Gu, 2024), represent a recent innovation, balancing the computational efficiency of recurrent networks with the expressive capacity of transformers. Mamba employs a selective state-space mechanism that efficiently captures temporal dependencies without the overhead of self-attention. Mamba-2 further improves long-horizon memory retention, making it competitive even in highly memory-dependent tasks. Recent work has integrated Mamba into decision-making frameworks, such as Decision Mamba (Lv et al., 2025; Ota, 2024), highlighting its adaptability across tasks.

These three architectural paradigms offer distinct strengths: recurrent networks provide robust solutions for short-term memory tasks, transformers excel in complex, long-horizon scenarios, and state-space models like Mamba achieve an optimal balance of computational efficiency and performance. Our work systematically benchmarks these architectures in RL, revealing their strengths, limitations, and suitable application domains.

## 2.2 BENCHMARKING NEURAL ARCHITECTURES IN RL

Recent years have seen increased interest in systematically evaluating different neural architectures across diverse RL tasks. Benchmark suites like POPGym (Morad et al., 2023) and Memory Gym (Pleines et al., 2024) have been developed specifically to assess how different memory models perform under varying degrees of partial observability. These benchmarks have revealed that no single architecture dominates across all tasks. While transformers excel when long but finite context is needed, recurrent networks often have an edge in continuously evolving tasks or when the agent must generalize from limited training data. Hybrid approaches that combine the strengths of different architectures (such as the Decision Mamba-Hybrid (Huang et al., 2024) that uses Mamba for long-term memory and a transformer for short-term decision-making) have shown particular promise in complex environments.

Our work provides a systematic comparison of PPO implementations across diverse neural architectures (MLP, LSTM, GRU, Transformer-XL, GTrXL, Mamba, and Mamba-2) within a unified framework. By focusing solely on these base architectures without external memory augmentation, we offer a clear analysis of their intrinsic memory capabilities, enabling a direct evaluation of the trade-offs in performance and computational efficiency across various environments.

## 3 BENCHMARKING SETTINGS

### 3.1 NEURAL NETWORK ARCHITECTURE IN RL

To ensure consistency and reproducibility, we build our implementations on CleanRL (Huang et al., 2022b), a widely used open-source library for RL algorithms. In our study, the core PPO algorithm is kept fixed while only the neural network architecture is varied, enabling fair comparisons that isolate architectural differences from algorithmic factors. Specifically, we benchmark the following neural network architectures:

(1) **MLP:** Standard multi-layer perceptron with ReLU activations, serving as our baseline. In the baseline configuration, agents received only the current observation at each timestep, referred to as *PPO-1* or *MLP Obs. Stack 1*. To introduce a simple memory mechanism, we experimented with *PPO-4*, also referred as *MLP Obs. Stack 4*, where four consecutive observations were concatenated and fed into the network. For Atari, this follows the standard practice of frame stacking the last four frames. (2) **LSTM and GRU:** Recurrent networks implemented following CleanRL's PPO-LSTM structure, using a single recurrent layer. While this single-layer approach is consistent with typical implementations, it may have limitations in highly memory-dependent tasks. (3) **Transformer-XL, GTrXL:** Implemented using CleanRL's episodic memory structure for hidden states, which maintains information across episode boundaries through its segmented recurrent mechanism. This implementation also employs post-transformer MLP layers. The gated version introduces a learned gating mechanism in place of traditional skip connections. (4) **Mamba/Mamba-2:** Integrated using the official implementation from the `mamba-ssm` repository. For Mamba, we employed an optimized training approach utilizing the selective scan mechanism without resetting at episode boundaries, offering computational advantages but potentially introducing state leakage between episodes. We incorporated post-model MLP layers and layer normalization.

Particularly, for all architectures, we adjusted network sizes to maintain approximately equal parameter counts (see Table 5 in Appendix A), ensuring that performance differences reflect architectural capabilities rather than capacity disparities. Moreover, we provide single-file implementations for each architecture variant. This strategy makes architectural differences explicit while maintaining consistent handling of environment interactions, data collection, and policy updates.

### 3.2 TRAINING PROTOCOLS

We adopted consistent training protocols across all environments, while adjusting hyperparameters to account for their varying complexity, as summarized in Table 9 and detailed further in Appendix A.

For all environments, we used the default maximum episode length as specified in their standard implementations. In MiniGrid tasks, we further increased the challenge by reducing the agent's

observation window from the default 7×7 to 3×3, making the environments more partially observable and thus more reliant on memory-based architectures.

### 3.3 HYPERPARAMETERS AND HARDWARE

We primarily follow CleanRL's default PPO hyperparameters, with key parameters shown in Table 10 from the Appendix A. The only notable exception was the learning rate for Mamba-based models, which was reduced following recommendations from recent literature (Luo et al., 2024) to enhance stability.

For **Mamba**, we used the default hyperparameter settings provided in the official implementation, as the available configuration space is relatively narrow and recent work suggests strong performance without extensive tuning.

For **Transformer-based models**, we utilized recommended hyperparameters for environments where prior work was available (e.g., MiniGrid, MemoryGym). In environments lacking specific benchmarks, we adjusted settings based on the environment's complexity, particularly whether it requires short-term or long-term memory.

All experiments were conducted on the same condition servers. Training throughput, inference latency, and memory usage were measured using PyTorch's built-in profiling tools. For more details, please see Appendix B.

## 4 RESULTS AND ANALYSIS

We evaluate the performance of each architecture across diverse environments with varying requirements for memory, continuous control, and discrete decision-making.

### 4.1 CONTINUOUS CONTROL TASKS

To systematically evaluate the impact of neural architectures on continuous control tasks, we conducted benchmarks on three popular MuJoCo environments: Walker2d-v4, HalfCheetah-v4, and Hopper-v4. These tasks present distinct challenges, ranging from stability-focused control (Hopper) to speed and efficiency (HalfCheetah). Our results demonstrate that the optimal architecture is highly task-dependent, reflecting the varying dynamics and control complexities of each environment.

> Findings:
> - Mamba has the worst performance across all environments, while Mamba-2 shows significant improvement, with performance comparable to LSTM and GRU while training 5× faster.
> - MLP performs well in most tasks, except for Hopper, where short memory capabilities are critical for maintaining stability.

Specifically, our analysis reveals that environments with simpler, smooth dynamics (like HalfCheetah) are effectively modeled by feed-forward architectures (MLP), which achieve high performance with strong sample efficiency. In contrast, environments with higher stability demands (like Hopper) benefit from recurrent models (GRU, LSTM), which effectively capture temporal dependencies critical for maintaining balance. Notably, in Walker2d, MLPs again perform competitively, highlighting the value of simplicity when the task dynamics are less chaotic.

Specifically, Figure 1 presents learning curves for the MuJoCo environments (Walker2d-v4, HalfCheetah-v4, and Hopper-v4), revealing distinct architectural advantages across different continuous control tasks: (1) In **Walker2d-v4**, MLP achieves the best performance (approximately 3250), demonstrating outstanding stability and good sample efficiency. GRU and Mamba-2 follow closely, reflecting solid temporal modeling abilities, while Transformer-XL reaches a lower asymptote. Original Mamba trails substantially, suggesting optimization instability or inadequacy for complex motor control. (2) In **HalfCheetah-v4**, LSTM and PPO-1 consistently outperform all others, showing the highest reward (approximately 3350). The smooth progression indicates robustness and effective temporal integration in continuous action spaces. Transformer-XL, GRU, and Mamba-2 achieve

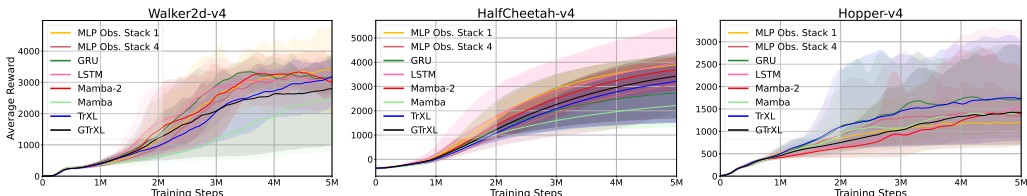

Figure 1: Average returns for MuJoCo tasks. MLP and LSTM demonstrate competitive or superior performance in Walker2d and HalfCheetah, while GRU and Transformer-XL perform best in Hopper.

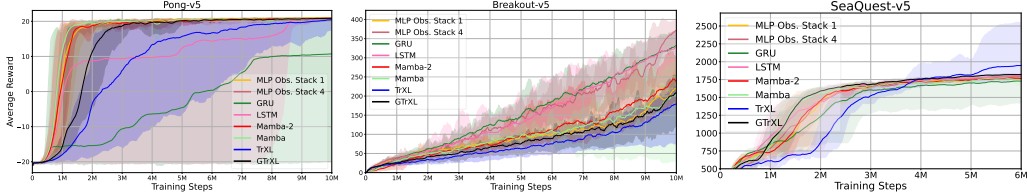

Figure 2: Average returns for Atari environments. Mamba and MLP with frame stacking excel in Pong, while LSTM and MLP with frame stacking perform best in Breakout. In SeaQuest, all architectures perform similarly, with only Transformer-XL slightly higher reward.

moderate success but lag in both final reward and convergence speed, while the original Mamba significantly underperforms. (3) In **Hopper-v4**, GRU and Transformer-XL perform best, reaching high and stable rewards, showing that moderate complexity sequence models suit tasks requiring balance and stability. MLP performs worse than others, reinforcing the value of recurrence in tasks with strong stability constraints.

## 4.2 DISCRETE CONTROL TASKS

Mamba and Mamba-2 excel in discrete control tasks like Pong, achieving rapid convergence due to efficient state-space modeling. However, Mamba struggles in more complex scenarios like Breakout, where its limited temporal modeling becomes a disadvantage. LSTM and GRU demonstrate strong performance in strategic environments like Breakout, but their added complexity can slow learning in simpler tasks (e.g. Pong). Gated Transformer-XL (GTrXL) shows only minor improvements over Transformer-XL, indicating that the gating mechanism provides limited benefits in these tasks.

> Finding: Mamba and Mamba-2 achieve fast convergence in reactive tasks like Pong but underperform in strategic environments like Breakout, where LSTM and GRU excel despite slower learning, while GTrXL offers only marginal gains over Transformer-XL.

For instance, results for Atari environments (Figure 2) reveal environment-specific architectural advantages: (1) In **Pong-v5**, Mamba, Mamba-2, and MLP rapidly reach maximum reward (20) with excellent sample efficiency (approximately 2M steps), indicating effectiveness in deterministic, reaction-time critical environments. GTrXL performs slightly better than standard Transformer-XL, suggesting that the gating mechanism enhances learning stability. LSTM and GRU exhibit slower learning, indicating that their recurrent nature introduces additional complexity in learning optimal behaviors, which may not be necessary for a straightforward, deterministic task like Pong. (2) For **Breakout-v5**, PPO-4 and recurrent architectures (GRU, LSTM) achieve the highest performance (approximately 360), steadily increasing reward and demonstrating good generalization and representation learning. All other architectures perform worse, implying difficulties with modeling task-specific structured temporal dynamics or input complexity in this Atari task.

## 4.3 PARTIALLY OBSERVABLE CONTROL TASKS

Traditional recurrent architectures (GRU/LSTM) excel in simpler, short-horizon partially observable tasks. Transformer-XL demonstrates strong performance in complex, partially observable settings, ef-

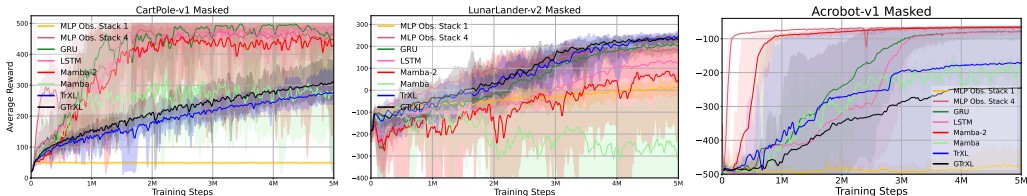

Figure 3: Average returns for masked classic control tasks. Recurrent architectures and stacked MLPs excel in CartPole, while Transformer-XL performs best in LunarLander.

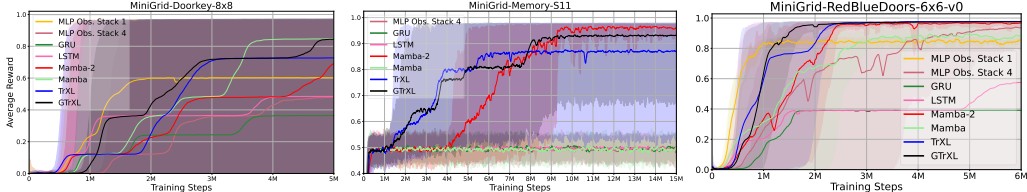

Figure 4: Average returns for MiniGrid environments. In DoorKey-8x8, original Mamba shows the fastest convergence, while in Memory-S11, only Transformer-XL and Mamba-2 achieve meaningful learning, with Mamba-2 reaching near-optimal performance.

fectively integrating information across longer time spans. GTrXL offers no substantial improvement over Transformer-XL, indicating limited benefits from the gating mechanism. Mamba exhibits poor performance and instability, potentially due to implementation challenges and information leakage across episodes. PPO-4 achieves fast and reliable performance, balancing simplicity and short-term memory.

> Finding: GRU and LSTM perform well in simple, short-horizon tasks, while Transformer-XL excels in complex, partially observable settings; GTrXL adds little benefit, Mamba struggles with instability, and PPO-4 offers fast, stable learning with minimal complexity.

For example, Figure 3 shows results for masked classic control environments (CartPole-v1, LunarLander-v2, and Acrobot-v1), where we removed all velocity information to create partial observability: (1) In **CartPole-v1 Masked**, GRU, LSTM, PPO-4, and Mamba-2 rapidly achieve maximum reward (approximately 500), demonstrating superior sample efficiency (approximately 1.5M steps) and robustness in masked, short-horizon tasks. (2) Transformer-XL has slower convergence and lower rewards, possibly due to overfitting or a lack of efficient representation of short-term masked inputs. (3) For **LunarLander-v2 Masked**, Transformer-XL/GTrXL, and PPO-4 are highly effective (200+ reward), especially Transformer-XL, which steadily overcomes partial observability (approximately 3M steps), revealing advantages of self-attention in inferring masked state elements. Mamba and Mamba-2 notably struggle, highlighting possible brittleness to masked or noisy observations. GRU and LSTM perform moderately well but plateau quickly, suggesting limitations in extracting masked information compared to self-attention mechanisms. (4) In **Acrobot-v1 Masked**, Mamba-2 and PPO-4 reach near-optimal performance and attain the best final return. LSTM and GRU eventually approach similar asymptotes but converge more slowly. Transformer-XL plateaus substantially lower, GTrXL is worse, and PPO-1 baselines fail to learn under masking. This suggests that for dynamics with strong temporal coupling, *selective state-space modeling* (Mamba-2) and classical recurrence are more effective than self-attention, while simple feed-forward memory proxies are insufficient.

## 4.4 MEMORY-INTENSIVE TASKS

Based on the comprehensive experiment results, Mamba-2, Transformer-XL, and GTrXL are the only architectures capable of effectively solving long-horizon memory tasks, with GTrXL demonstrating more stable learning curves due to the gating mechanism. Both Mamba variants and Transformer-XL also excel in environments requiring complex credit assignment, such as DoorKey. In contrast,

conventional recurrent architectures (LSTM, GRU) and simple MLP models struggle in these settings. For example, as shown in Figure 4 presents results for our most memory-demanding environments: MiniGrid Memory-S11, DoorKey-8x8, and RedBlueDoors-6x6.

> **Finding**: Mamba-2, Transformer-XL, and GTrXL are uniquely effective in long-horizon memory and complex credit assignment tasks, with GTrXL offering added stability via gating, while traditional recurrent and MLP models fall short.

In **DoorKey-8x8**, the original Mamba architecture demonstrates remarkable sample efficiency, rapidly converging to the highest reward. Its learning curve sharply surpasses other models, indicating superior mid-length memory and efficient representation learning. Transformer-XL and GTrXL achieve a strong result slightly slower, demonstrating good generalizability but moderate sample efficiency. Mamba-2 achieves moderate success more slowly, suggesting potential optimization challenges on shorter-horizon memory tasks compared to original Mamba. Surprisingly, a simple PPO-1 outperforms standard recurrent architectures (LSTM, GRU), suggesting that minimal state abstraction is sufficient in less complex tasks.

In the highly challenging **Memory-S11** environment, Mamba-2 significantly outperforms all other architectures, achieving near-optimal reward (approximately 0.96) with remarkable stability. Transformer-XL exhibits steady and stable performance but converges at a slightly lower optimal reward with higher variance. GTrXL achieves similar results but get higher mean reward, indicating that gating mechanism provides more stable learning curves. LSTM, GRU, MLP, and original Mamba show no meaningful learning beyond random exploration, indicating their limited capabilities in tasks requiring extensive memory.

In **RedBlueDoors-6×6**, the agent is required to remember a color cue and perform delayed, symbol-conditioned navigation. TrXL and GTrXL solve it reliably. Mamba-2 also solves the task but exhibits small oscillations during training. The original Mamba plateaus lower.

## 4.5 COMPUTATIONAL EFFICIENCY ANALYSIS

The detailed measurements of computational efficiency across architectures are presented in the Appendix A, while the average results across tasks are summarized in Table 1. These metrics are critical for evaluating the practical applicability of each approach in resource-constrained settings:

Table 1: Average Computational Metrics Across Architectures

| Metric | PPO-1 | PPO-4 | LSTM | GRU | TrXL | GTrXL | Mamba | Mamba-2 |
|---|---|---|---|---|---|---|---|---|
| Final Steps Per Second (SPS) ↑ | **3539** | 3305 | 604 | 701 | 1856 | 1890 | 2734 | 2455 |
| Training Time (min) ↓ | **16.59** | 18.84 | 121.90 | 91.04 | 30.33 | 29.42 | 21.20 | 22.97 |
| Inference Latency (ms) ↓ | **0.856** | 0.899 | 1.006 | 0.971 | 2.171 | 2.147 | 1.304 | 1.489 |
| GPU Mem. Allocated (GB) ↓ | **0.035** | 0.660 | 0.194 | 0.194 | 1.765 | 1.330 | 0.217 | 0.219 |
| GPU Mem. Reserved (GB) ↓ | **0.327** | 0.983 | 0.343 | 0.349 | 5.508 | 4.968 | 0.362 | 0.662 |

> **Finding**: Mamba models are ideal for resource-constrained environments where fast throughput and low memory usage are critical compared with other architectures such as Transformer XL, LSTM and GRU.

Mamba achieves exceptional computational efficiency, being 4.5× faster than LSTM, 3.9× faster than GRU, and 1.5× faster than Transformer-XL, while maintaining low memory usage (8× less than Transformer-XL). Mamba-2, despite being slightly slower, retains significant efficiency advantages over LSTM and Transformer-XL.

Mamba achieves an average of 2734 *steps per second (SPS)*, which is significantly faster than LSTM, GRU, and Transformer-XL, though still slower than MLP (1.3×). This throughput advantage translates directly to training time improvements, with Mamba completing the same number of environment interactions in approximately one-quarter the time required by LSTM.

In terms of *inference latency* (reported in the Appendix), Mamba maintains relatively low response times (1.30 ms on average), comparable to recurrent models such as LSTM (1.01 ms) and GRU

(0.971 ms). While Mamba is approximately 1.3× slower than GRU and LSTM, it is still significantly faster (1.66×) than Transformer-XL, which averages 2.17 ms. Despite these differences, all models demonstrate low-latency performance overall, and such variations are unlikely to pose significant challenges in most real-world applications where fast decision-making is required.

*Memory efficiency* shows perhaps the most dramatic differences between architectures. Mamba requires only 0.217 GB of GPU memory on average, which is 8.1× less than Transformer-XL (1.765 GB) while achieving comparable or superior performance in most environments. This substantial memory advantage makes Mamba suitable for deployment on resource-constrained edge devices or for scaling to larger batch sizes on standard hardware.

## 4.6 ARCHITECTURE-ENVIRONMENT COMPATIBILITY

Our comprehensive evaluation reveals clear patterns regarding which architectures excel in particular environments:

**Memory-independent tasks:** In environments with relatively smooth or Markovian dynamics, such as continuous control tasks (Walker2d, HalfCheetah) and reaction-based games (Pong), simpler architectures like MLPs and Mamba perform effectively by capturing immediate dependencies with high stability. However, for tasks requiring strategic planning (e.g., Breakout), models with recurrent structure or stacked inputs (like LSTM and MLP Stack-4) are better suited.

**Partially observable environments:** The optimal architecture depends on the complexity of the hidden state. In simpler masked tasks (CartPole), traditional recurrent architectures (GRU/LSTM) excel, while more complex partially observable environments (LunarLander) benefit from attention mechanisms (Transformer-XL) that can more effectively infer hidden variables.

**Memory tasks:** In environments requiring moderate memory capabilities (DoorKey-8x8), the original Mamba architecture demonstrates outstanding sample efficiency and performance, suggesting its selective state-space approach provides an ideal inductive bias for mid-length memory requirements. However in long-horizon tasks only Mamba-2 and Transformer-XL can effectively solve tasks requiring extensive memory (Memory-S11), with Mamba-2 achieving superior performance. This indicates that advanced state-space models with selective attention mechanisms are uniquely suited to long-term dependency modeling in RL.

These patterns provide actionable guidance for practitioners: architecture selection should be driven by the specific memory and control requirements of the target environment, with simpler architectures preferred unless the task specifically demands long-term memory retention or complex partially observable state inference.

## 4.7 PRACTICAL GUIDELINES FOR PRACTITIONERS

Based on our comprehensive evaluation, we propose the following guidelines for selecting neural architectures in RL:

(1) **Start with MLP**: For most tasks, particularly those with largely Markovian dynamics (e.g., MuJoCo), Multi-Layer Perceptrons (MLPs) provide an excellent balance of performance, stability, and computational efficiency. They are fast to train and offer strong baseline performance.

(2) **Prioritize Mamba-2 for Sequence Tasks**: If the task involves temporal dependencies or partial observability, Mamba-2 should be your first choice. It offers a unique combination of fast training (approximately 5× faster than LSTM/GRU) and competitive performance, making it a practical first option for sequence modeling.

(3) **Explore LSTM and GRU if Mamba-2 Falls Short**: In cases where Mamba-2 does not achieve satisfactory results, consider trying LSTM and GRU. While they require significantly longer training times, they may outperform Mamba-2 in some environments due to their well-established memory modeling capabilities.

(4) **Reserve Transformers for Challenging Memory Tasks**: Transformers (Transformer-XL, GTrXL) should be considered only for environments that are extremely memory-intensive, such as long-horizon planning or complex partially observable tasks. Their high computational cost and

implementation complexity make them unsuitable for most practical applications unless the task is specifically designed to benefit from long-range memory modeling.

These guidelines serve as a starting point, but optimal architecture selection should ultimately depend on the specific characteristics of the target environment and the available computational resources.

### 4.8 FUTURE WORK

This work opens several directions for further research. First, a more thorough hyperparameter optimization process, beyond default CleanRL settings, could provide a fairer comparison across architectures, particularly for Transformer-XL and Mamba, which may benefit from task-specific tuning. Second, architectural ablation studies are needed to isolate the contributions of specific components, such as recurrence depth, attention heads, or state-space scan mechanisms, to better understand performance–efficiency tradeoffs.

Another important technical improvement involves adding proper hidden state resets to Mamba, as our current implementation allows information leakage between episodes, potentially affecting performance in episodic tasks. Future work should also explore deeper or stacked versions of each architecture to investigate scaling behaviors. Finally, extending our evaluation beyond PPO to other RL algorithms (e.g., SAC (Haarnoja et al., 2018), TD3 (Fujimoto et al., 2018)) and exploring hybrid architectures that combine recurrence, attention, and state-space memory could lead to more flexible and robust solutions for partially observable environments.

## 5 DISCUSSION AND CONCLUSION

In this work, we systematically evaluate neural network architectures for RL under a unified PPO framework. We find that greater complexity does not necessarily yield better performance: MLPs remain strong in fully observable tasks, Mamba-2 combines high throughput with competitive accuracy in memory-intensive settings, and Transformers excel only in extreme memory-demanding tasks at much higher cost. These results highlight practical trade-offs and provide guidance for architecture selection in RL:

1) **Mamba vs. Mamba-2:** While Mamba exhibits inconsistent performance across environments, Mamba-2 consistently achieves strong results. It combines competitive accuracy in memory-intensive tasks with outstanding efficiency, training $5.3\times$ faster than LSTM and GRU and requiring $8.1\times$ less GPU memory than Transformer-XL.

2) **MLPs for Markovian tasks:** Simpler architectures such as MLPs remain highly effective in fully observable environments (e.g., MuJoCo). Their low complexity and fast training make them a practical first choice when temporal dependencies are limited.

3) **Efficient recurrent modeling:** For tasks with moderate temporal dependencies, Mamba-2 is a strong starting point due to its efficient state-space design and high throughput. If Mamba-2 proves unstable or underperforms, LSTM and GRU remain robust alternatives.

4) **Transformers for extreme memory demands:** Transformer-based models (e.g., Transformer-XL, GTrXL) are computationally expensive and memory-intensive. They are best suited for environments with extensive memory requirements (e.g., MiniGrid Memory-S11). Between them, GTrXL generally offers smoother learning curves, lower variance, and slightly stronger final performance.

5) **Complexity vs. performance:** Crucially, our findings challenge the assumption that more complex architectures inherently yield better performance. The strong results of MLPs and Mamba-2 demonstrate that efficient architectures are often preferable, particularly in resource-constrained settings. We recommend practitioners begin with lightweight models and scale complexity only when task demands justify it.

Overall, these findings highlight the trade-offs among efficiency, stability, and representational capacity across architectures. For practical applications, we recommend starting with efficient models such as MLPs or Mamba-2, and scaling to more complex architectures like LSTMs, GRUs, or Transformers only when task characteristics necessitate it.

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

## A PERFORMANCE METRICS.

Table 2: Final Steps Per Second (SPS) for Various Architectures and Environments (Rounded)

| Environment | PPO-1 | PPO-4 | LSTM | GRU | TrXL | GTrXL | Mamba | Mamba-2 |
|---|---|---|---|---|---|---|---|---|
| MiniGrid-MemoryS11-v0 | 3191 | 2544 | 802 | 924 | 1697 | 1869 | 2850 | 2698 |
| MiniGrid-DoorKey-8x8-v0 | 3496 | 2770 | 828 | 957 | 1684 | 1868 | 3031 | 2844 |
| Breakout-v5 | 1521 | 1487 | 626 | 701 | 1188 | 1179 | 1329 | 1245 |
| Pong-v5 | 1753 | 1699 | 666 | 742 | 1322 | 1310 | 1512 | 1404 |
| CartPole-v1 | 7046 | 6930 | 991 | 1151 | 3781 | 3594 | 4643 | 3827 |
| LunarLander-v2 | 5979 | 5726 | 897 | 1061 | 2585 | 2489 | 4018 | 3398 |
| Walker2d-v4 | 2738 | 2675 | 210 | 256 | 1430 | 1436 | 2270 | 2121 |
| HalfCheetah-v4 | 3315 | 3212 | 211 | 259 | 1577 | 1706 | 2632 | 2429 |
| Hopper-v4 | 2808 | 2703 | 208 | 256 | 1440 | 1555 | 2325 | 2133 |
| **Average** | **3539** | 3305 | 604 | 701 | 1856 | 1890 | 2734 | 2455 |

Table 3: Training Time (in Minutes) for Various Architectures and Environments (3 million total timesteps)

| Environment | PPO-1 | PPO-4 | LSTM | GRU | TrXL | GTrXL | Mamba | Mamba-2 |
|---|---|---|---|---|---|---|---|---|
| MiniGrid-DoorKey-8x8-v0 | 14.31 | 18.06 | 60.33 | 52.22 | 29.68 | 26.76 | 16.50 | 17.59 |
| MiniGrid-MemoryS11-v0 | 15.67 | 19.66 | 62.31 | 54.04 | 29.46 | 26.74 | 17.75 | 18.54 |
| Breakout-v5 | 32.91 | 33.65 | 79.78 | 71.29 | 42.10 | 42.43 | 37.66 | 40.17 |
| Pong-v5 | 28.55 | 29.46 | 75.03 | 67.34 | 37.83 | 38.20 | 33.10 | 35.64 |
| CartPole-v1 | 7.11 | 7.23 | 50.41 | 43.43 | 13.23 | 13.92 | 10.78 | 13.07 |
| LunarLander-v2 | 8.37 | 8.74 | 55.66 | 47.11 | 19.34 | 20.10 | 12.45 | 14.72 |
| Walker2d-v4 | 18.27 | 18.70 | 237.78 | 194.69 | 34.95 | 35.13 | 22.03 | 23.58 |
| HalfCheetah-v4 | 15.10 | 15.58 | 236.42 | 192.39 | 31.70 | 29.32 | 19.00 | 20.59 |
| Hopper-v4 | 17.82 | 18.51 | 239.37 | 96.81 | 34.70 | 32.15 | 21.51 | 23.44 |
| **Average** | **16.59** | 18.84 | 121.90 | 91.04 | 30.33 | 29.42 | 21.20 | 22.97 |

Table 4: Evaluation Results (Mean ± Std) of Final Average Episode Return.

| Environment | PPO-1 | PPO-4 | LSTM | GRU | TrXL | GTrXL | Mamba | Mamba-2 |
|---|---|---|---|---|---|---|---|---|
| **MiniGrid** | | | | | | | | |
| MemoryS11 | – | 0.49 ± 0.02 | 0.51 ± 0.04 | 0.49 ± 0.02 | 0.88 ± 0.18 | 0.93 ± 0.11 | 0.49 ± 0.03 | **0.96 ± 0.01** |
| DoorKey | 0.60 ± 0.50 | 0.48 ± 0.51 | 0.49 ± 0.52 | 0.36 ± 0.50 | 0.73 ± 0.45 | 0.84 ± 0.34 | **0.85 ± 0.34** | 0.69 ± 0.43 |
| **Atari** | | | | | | | | |
| Breakout | 220.5 ± 27.6 | **372.7 ± 20.3** | 327.8 ± 38.4 | 332.2 ± 43.4 | 180.4 ± 55.5 | 208.7 ± 38.3 | 202.9 ± 89.0 | 239.2 ± 39.5 |
| Pong | **20.98 ± 0.03** | 20.69 ± 0.33 | 20.75 ± 0.14 | 10.71 ± 18.76 | 20.41 ± 1.20 | 20.81 ± 0.1 | 20.82 ± 0.10 | 20.89 ± 0.09 |
| **Classic Control** | | | | | | | | |
| Cartpole | 49.4 ± 2.1 | 458.7 ± 28.9 | **484.6 ± 26.6** | 449.4 ± 152.8 | 281.8 ± 41.2 | 305.8 ± 40.6 | 272.8 ± 104.7 | 434.2 ± 90.5 |
| LunarLander | 16.6 ± 28.9 | 190.1 ± 11.8 | 123.8 ± 38.0 | 204.8 ± 23.2 | **244.9 ± 13.1** | 236.5 ± 8.5 | -313.2 ± 235.1 | 42.5 ± 182.2 |
| **MuJoCo** | | | | | | | | |
| HalfCheetah | 3918.8 ± 312.8 | 3116.4 ± 612.0 | **3997.4 ± 1378.8** | 2745.0 ± 1118.1 | 3240.8 ± 848.8 | 3464.0 ± 921.7 | 2238.7 ± 627.2 | 3718.0 ± 528.5 |
| Hopper | 1194.1 ± 200.4 | 1366.4 ± 495.1 | 1574.4 ± 842.7 | 1676.6 ± 724.5 | **1712.0 ± 741.6** | 1428.5 ± 711.4 | 1409.2 ± 635.7 | 1390.2 ± 639.3 |
| Walker2d | **3379.3 ± 1039.8** | 3152.1 ± 193.0 | 3056.1 ± 547.4 | 3170.3 ± 355.4 | 3206.1 ± 528.0 | 2765.7 ± 937.4 | 2512.6 ± 1000.5 | 3038.0 ± 505.2 |

Table 5: Number of Parameters (in Thousands) for Various Architectures and Environments

| Environment | PPO-1 | LSTM | GRU | TrXL | GTrXL | Mamba | Mamba-2 |
|---|---|---|---|---|---|---|---|
| Hopper-v4 | 39.7 | 38.6 | 39.1 | 37.8 | 44.9 | 40.1 | 43.8 |
| HalfCheetah-v4 | 40.4 | 39.2 | 39.7 | 38.3 | 45.5 | 40.7 | 44.4 |
| Walker2d-v4 | 40.4 | 39.2 | 39.7 | 38.3 | 45.5 | 40.7 | 44.4 |
| Pong-v5 | 2527.2 | 2468.8 | 2271.7 | 2669.2 | 2639.3 | 2413.9 | 2805.3 |
| Breakout-v5 | 2731.2 | 2468.3 | 2271.1 | 2668.2 | 2638.4 | 2413.0 | 2804.3 |
| LunarLander-v2 | 1057.8 | 1057.8 | 1042.8 | 1035.3 | 1070.1 | 1089.0 | 973.0 |
| CartPole-v1 | 265.2 | 265.5 | 268.6 | 261.9 | 264.5 | 262.6 | 226.0 |
| MiniGrid-MemoryS11-v0 | 2470.9 | 2473.1 | 2276.0 | 2408.4 | 2591.7 | 2470.9 | 2426.0 |
| MiniGrid-DoorKey-8x8-v0 | 2531.8 | 2473.1 | 2465.2 | 2408.4 | 2510.6 | 2357.8 | 2426.0 |

Table 6: Inference Latency (ms) for Various Architectures and Environments

| Environment | PPO-1 | PPO-4 | LSTM | GRU | TrXL | GTrXL | Mamba | Mamba-2 |
|---|---|---|---|---|---|---|---|---|
| MiniGrid-MemoryS11-v0 | 1.139 | 1.309 | 1.290 | 1.254 | 2.917 | 2.729 | 1.603 | 1.780 |
| MiniGrid-DoorKey-8x8-v0 | 1.093 | 1.249 | 1.215 | 1.171 | 2.845 | 2.666 | 1.534 | 1.754 |
| Breakout-v5 | 1.016 | 1.040 | 1.180 | 1.127 | 1.792 | 1.941 | 1.495 | 1.668 |
| Pong-v5 | 1.024 | 1.047 | 1.167 | 1.132 | 1.775 | 1.946 | 1.488 | 1.690 |
| CartPole-v1 | 0.739 | 0.729 | 0.909 | 0.887 | 1.455 | 1.600 | 1.158 | 1.347 |
| LunarLander-v2 | 0.725 | 0.736 | 0.956 | 0.920 | 1.973 | 2.249 | 1.176 | 1.367 |
| Walker2d-v4 | 0.658 | 0.660 | 0.776 | 0.748 | 2.268 | 2.073 | 1.093 | 1.260 |
| HalfCheetah-v4 | 0.649 | 0.656 | 0.780 | 0.747 | 2.245 | 2.056 | 1.099 | 1.263 |
| Hopper-v4 | 0.659 | 0.666 | 0.780 | 0.749 | 2.272 | 2.061 | 1.093 | 1.272 |
| **Average** | **0.856** | 0.899 | 1.006 | 0.971 | 2.171 | 2.147 | 1.304 | 1.489 |

Table 7: GPU Memory Allocated (GB) for Various Architectures and Environments

| Environment | PPO-1 | PPO-4 | LSTM | GRU | TrXL | GTrXL | Mamba | Mamba-2 |
|---|---|---|---|---|---|---|---|---|
| MiniGrid-MemoryS11-v0 | 0.702 | 2.644 | 0.705 | 0.701 | 5.729 | 4.371 | 0.788 | 0.799 |
| MiniGrid-DoorKey-8x8-v0 | 0.702 | 2.644 | 0.705 | 0.705 | 2.341 | 1.870 | 0.796 | 0.799 |
| Pong-v5 | 0.113 | 0.274 | 0.116 | 0.115 | 0.712 | 0.636 | 0.125 | 0.125 |
| Breakout-v5 | 0.113 | 0.274 | 0.116 | 0.116 | 0.730 | 0.650 | 0.125 | 0.125 |
| CartPole-v1 | 0.020 | 0.020 | 0.021 | 0.021 | 0.353 | 0.282 | 0.023 | 0.025 |
| LunarLander-v2 | 0.020 | 0.020 | 0.036 | 0.035 | 1.034 | 0.804 | 0.039 | 0.043 |
| Walker2d-v4 | 0.018 | 0.022 | 0.018 | 0.018 | 1.657 | 1.125 | 0.020 | 0.020 |
| HalfCheetah-v4 | 0.018 | 0.022 | 0.018 | 0.018 | 1.648 | 1.108 | 0.020 | 0.020 |
| Hopper-v4 | 0.018 | 0.021 | 0.018 | 0.018 | 1.677 | 1.128 | 0.019 | 0.019 |
| **Average** | **0.035** | 0.660 | 0.194 | 0.194 | 1.765 | 1.330 | 0.217 | 0.219 |

Table 8: GPU Memory Reserved (GB) for Various Architectures and Environments

| Environment | PPO-1 | PPO-4 | LSTM | GRU | TrXL | GTrXL | Mamba | Mamba-2 |
|---|---|---|---|---|---|---|---|---|
| MiniGrid-MemoryS11-v0 | 1.133 | 3.941 | 1.131 | 1.203 | 18.630 | 15.707 | 1.236 | 1.285 |
| MiniGrid-DoorKey-8x8-v0 | 1.133 | 3.941 | 1.131 | 1.127 | 8.494 | 12.556 | 1.238 | 1.285 |
| Pong-v5 | 0.264 | 0.393 | 0.273 | 0.268 | 2.002 | 1.781 | 0.285 | 0.396 |
| Breakout-v5 | 0.264 | 0.391 | 0.273 | 0.268 | 2.204 | 1.984 | 0.283 | 0.396 |
| CartPole-v1 | 0.027 | 0.027 | 0.049 | 0.051 | 1.089 | 0.970 | 0.051 | 0.512 |
| LunarLander-v2 | 0.027 | 0.027 | 0.068 | 0.066 | 2.955 | 2.292 | 0.063 | 0.541 |
| Walker2d-v4 | 0.031 | 0.051 | 0.053 | 0.053 | 5.461 | 3.631 | 0.033 | 0.514 |
| HalfCheetah-v4 | 0.031 | 0.051 | 0.053 | 0.053 | 3.668 | 2.504 | 0.033 | 0.514 |
| Hopper-v4 | 0.031 | 0.029 | 0.053 | 0.053 | 5.070 | 3.288 | 0.033 | 0.514 |
| **Average** | **0.327** | 0.983 | 0.343 | 0.349 | 5.508 | 4.968 | 0.362 | 0.662 |

# B HYPERPARAMETERS

All experiments were conducted on a single NVIDIA RTX A5000 GPU and Intel Xeon W-1390p to ensure consistent performance measurement. For reproducibility and transparency, we list the detailed hyperparameters and training settings[1] used throughout our experimental evaluation:

---

[1] https://anonymous.4open.science/r/BenchNetRL-A718

Table 9: Training settings. Steps and seeds to the right are for computational metrics.

| Domain | Environment | Steps | Seeds | *Steps* | *Seeds* |
|--------|-------------|-------|-------|---------|---------|
| Classic Control | CartPole-v1 | 5M | 8 | *3M* | *3* |
| Classic Control | LunarLander-v2 | 5M | 8 | *3M* | *3* |
| Classic Control | Acrobot-v1 | 5M | 8 | *3M* | *3* |
| Atari | Pong-v5 | 10M | 8 | *3M* | *3* |
| Atari | Breakout-v5 | 10M | 8 | *3M* | *3* |
| Atari | SeaQuest-v5 | 6M | 8 | *3M* | *3* |
| Mujoco | HalfCheetah-v4 | 5M | 8 | *3M* | *3* |
| Mujoco | Hopper-v4 | 5M | 8 | *3M* | *3* |
| Mujoco | Walker2d-v4 | 5M | 8 | *3M* | *3* |
| MiniGrid | MemoryS11-v0 | 15M | 8 | *3M* | *3* |
| MiniGrid | DoorKey-8x8-v0 | 5M | 8 | *3M* | *3* |
| MiniGrid | RedBlueDoors-6x6-v0 | 6M | 8 | *3M* | *3* |

Table 10: Common Training Hyperparameters by Domain

| Parameter | MiniGrid | MuJoCo | Classic Control | Atari |
|-----------|----------|--------|-----------------|-------|
| Total timesteps | $1.5 \times 10^{7}$ | $5 \times 10^{6}$ | $5 \times 10^{6}$ | $1 \times 10^{7}$ |
| Batch size | 8 192 | 16 384 | 2 048 | 2 048 |
| Mini-batch size | 1 024 | 2 048 | 256 | 256 |
| # Environments | 16 | 8 | 16 | 16 |
| # Steps / env | 512 | 2 048 | 128 | 128 |
| Update epochs | 4 | 10 | 4 | 4 |
| Discount factor ($\gamma$) | 0.995 | 0.99 | 0.99 | 0.99 |
| GAE $\lambda$ | 0.95 | 0.95 | 0.95 | 0.95 |
| Learning rate (Adam) | $2.5 \times 10^{-4}$ | $3 \times 10^{-4}$ | $2.5 \times 10^{-4}$ | $2.5 \times 10^{-4}$ |
| Value loss coef. ($c_v$) | 0.5 | 0.5 | 0.5 | 0.5 |
| Clip coefficient | 0.1 | 0.1 | 0.1 | 0.1 |
| Max grad norm | 0.5 | 0.5 | 0.5 | 0.5 |

Table 11: Model-Specific Hyperparameters (values listed as MiniGrid / MuJoCo / Classic / Atari)

| Param | PPO-4 / PPO-1 | LSTM | GRU | TRXL | GTrXL | Mamba | Mamba-2 |
|-------|---------------|------|-----|------|-------|-------|---------|
| Hidden dim | 512/90/256/512 | 512/64/256/512 | 512/64/256/512 | 384/64/284/512 | 376/64/336/448 | 380/70/284/450 | 384/64/164/512 |
| Entropy coef. | 1e-4/0/0.01/0.01 | 1e-4/0/0.01/0.01 | 1e-4/0/0.01/0.01 | 1e-2/0.01/0.01/0.01 | 0.01/0.01/0.01/0.01 | 1e-2/0/0.01/0.01 | 1e-2/0/0.01/0.01 |
| RNN hidden dim | – | 256/64/128/256 | 256/78/160/256 | – | – | – | – |
| TRXL layers | – | – | – | 3/3/1/1 | 2/2/2/1 | – | – |
| TRXL heads | – | – | – | 4/4/4/2 | 4/4/4/2 | – | – |
| TRXL memory len. | – | – | – | 119/64/32/64 | 119/64/32/64 | – | – |
| $d_{\text{state}}$ | – | – | – | – | – | 64/64/64/64 | 128/64/128/64 |
| $d_{\text{conv}}$ | – | – | – | – | – | 4/4/4/4 | 4/4/4/4 |
| Expand ratio | – | – | – | – | – | 2/1/2/1 | 2/2/2/1 |
| Learning rate | – | – | – | – | – | 1.5e-4/3e-4/1.5e-4/1.5e-4 | 1.5e-4/3e-4/1.5e-4/1.5e-4 |

LICENSES FOR EXISTING ASSETS

We use the following open-source assets, with licenses listed for each:

- **gym-minigrid (MiniGrid)**: Apache License 2.0.
- **MuJoCo physics engine**: Apache License 2.0; **mujoco-py** bindings: MIT License.
- **Arcade Learning Environment (Atari ALE)**: GNU General Public License v3.0.
- **CleanRL**: MIT License.
- **mamba-ssm**: MIT License.

