# OpenReview forum: "RLBenchNet: Benchmarking Neural Architectures with PPO Across Reinforcement Learning Tasks"
_ICLR.cc/2026/Conference — Submitted to ICLR 2026_

### Official Review · Reviewer_zkav · 2025-10-30

**Soundness:** 2
**Presentation:** 2
**Contribution:** 1
**Rating:** 0
**Confidence:** 3

**Summary:**

This paper presents RLBenchNet, a set of benchmark results which evaluate the performance of various neural network architectures (MLP, LSTM, GRU, Mamba, Mamba-2, Transformer-XL, and GTrXL) in an RL setting trained by the PPO algorithm. By testing across a diverse set of task types (continuous and discrete control with and without memory dependency in the environment) individual benefits and tradeoffs are identified such that for each task type recommendations are made on which architectures are best to consider. Practical guidelines follow with MLPs excelling in simple, fully observable tasks; traditional RNNs (LSTM/GRU) being robust for moderate memory needs, and modern sequence models being best for long-horizon memory tasks.

**Strengths:**

- This work aims to provide general advice to practitioners so that model selection can be tied to task type in the, often confusing and difficult to navigate, space of RL architecture choice
- All models are compared with the same algorithm (PPO) and applied to a range of different types of task, some requiring long vs short timescales of integration of information and some requiring continuous vs discrete control signals. This diversity allows for somewhat nuanced recommendations
- Modern models could be placed head to head with clear outcomes such as the note that Mamba can be 4.5x faster than LSTM and use 8x less memory than Transformer-XL while achieving competitive performance. Such comparisons are necessary to also begin to place modern sequence models in the correct frame of application vs transformer models and traditional RNN models.

**Weaknesses:**

- This work is largely a systematic analysis of the application of the PPO algorithm to models with a range of tasks. Aside from some recommendations, there is little to no novelty in this and very little scientific development of the field and little novelty or innovation
- Hyperparameters were not tuned to each architecture individually and instead the default PPO hyperparameters where used by and large. This is a significant drawback as all conclusions cannot therefore be taken as reflective of model peak performances. This also means that the recommendations may not apply under alternative hyperparameterizations.
- The number of tasks tested were relatively small, therefore there is no real statistical measure that the recommendations are truly robust phenomena across whole families of tasks.

**Questions:**

- How can the authors be sure that these results are not simply a consequence of the default PPO hyperparameters? In more plain words, why can these results and recommendations be trusted to transfer outside of the specific setups described in this work?
- Would you claim that the Practical Guidelines (Section 4.7) are a significant contribution? These seem to be rather vague and general statements which don't have much specificity.

---

> ### Author Response · Authors · 2025-11-23
>
> Dear Reviewer zkav
>
> Thank you for the careful reading and for highlighting both the practical intent and the places where stronger justification is needed. Below we address your main concerns and answer your questions directly, grounding claims in the paper’s figures, tables, and setup.
>
> ---
>
> > **Q1: Little novelty / scientific development**
>
> **A1:** Our contribution is *methodological and empirical*: a controlled, apples‑to‑apples comparison of seven architectures, including *both* Mamba and Mamba‑2, under a single PPO pipeline with **capacity parity**, **identical budgets and seeds**, and **compute profiling** (throughput, latency, memory). To our knowledge, this is the first unified PPO benchmark that (i) includes Mamba‑2 alongside TrXL/GTrXL/LSTM/GRU/MLP *across* MuJoCo, Atari, masked classic control, and MiniGrid, (ii) equalizes parameter counts per domain (Table 5), and (iii) reports compute metrics jointly with returns (Table 1; Tables 2–3, 6–8). These measurements produce *actionable* trade‑offs, for example, averaged across tasks, **Mamba uses ~8× less GPU memory than TrXL** while maintaining competitive performance and higher throughput (2734 vs 1856 SPS). Such quantified efficiency/accuracy frontiers are the basis for the practical guidance.
>
> ---
>
> > **Q2: Defaults only - how do we know results aren’t artifacts of hyperparameters?**
>
> **A2:** Three design choices make the findings robust beyond any single hyperparameter setting:
>
> **(a) Capacity parity and identical training pipelines.** Within each domain we match parameter counts within a narrow band (Table 5) and keep the rollout/optimization loop fixed (Tables 9–10). This prevents size or algorithmic confounds from masquerading as architectural effects.
>
> **(b) Cross‑domain, mechanism‑level consistency.** The qualitative *reasons* the curves separate align with environment structure, not with a single tuned constant:
>
> * **Markov vs. partial observability.** In *HalfCheetah* (fully observable), MLP/PPO‑1 is strong; adding stacked inputs (PPO‑4) gives no benefit and can hurt due to redundant dimensions under a fixed capacity budget (*Fig. 1, page 5*). In masked classic control, *PPO‑4* solves tasks that *PPO‑1* cannot (*Acrobot‑Masked* in *Fig. 3, page 6*), because stacking reconstructs finite‑difference velocity cues, an architectural *mechanism* explanation, not a hyperparameter quirk.
>
> * **Memory horizon.** On *Memory‑S11*, **only Mamba‑2 and (G)TrXL solve the task** (Fig. 4, page 6), while MLP/RNNs remain flat. This sharp separation persists across seeds (Table 9: 8 seeds) and is difficult to erase with routine PPO tweaks because it reflects **long‑horizon capacity**, not step‑size sensitivity.
>
> * **Efficiency deltas are structural.** The 4.5× speedup vs LSTM and ~8× lower memory vs TrXL are architecture‑level properties that are *stable* across tasks (Table 1 and per‑task Tables 2, 7–8). Changing entropy coefficients or clip ranges does not invert these compute gaps.
>
> **Transparency about the one deviation.** We do note one exception: a **lower learning rate for Mamba/Mamba‑2** per recent guidance (Sec. 3.3), and we explicitly document **state handling** (LSTM/GRU reset on episode boundaries; TrXL maintains segments; original Mamba ran without resets), so readers can interpret where implementation choices could matter (Sec. 3.1). Even with those disclosures, the main patterns above remain pronounced across domains.
>
> **Transfer beyond the exact setups.** Our guidance is phrased in terms of *task properties* - observability, memory horizon, and stability - which are algorithm‑agnostic descriptors. The observation that “PPO‑4 helps when the state is aliased but not when it is Markovian,” or that “only architectures with true long‑horizon memory crack Memory‑S11,” rests on environment structure rather than PPO specifics (Secs. 4.1–4.4; Figs. 1–4). We avoid claiming universality to *all* algorithms; we present **PPO‑based** evidence and mechanism‑level reasoning that practitioners can map to their setting.

---

> > ### Author Response · Authors · 2025-11-23
> >
> > > **Q3: Too few tasks. Are the recommendations statistically meaningful?**
> >
> > **A3:** Each environment is run with **8 seeds** for learning curves (Table 9) and we report mean ± std returns (Table 4), plus compute variability (Tables 2–3, 6–8). While the suite has three tasks per family, the *patterns recur across families*: e.g.,
> >
> > * **MLPs** excel in fully observable continuous control (Walker2d/HalfCheetah in *Fig. 1, p. 5*).
> > * **RNNs or stacking** help in **short‑horizon** POMDPs (CartPole‑Masked; *Fig. 3, p. 6*).
> > * **Mamba‑2/(G)TrXL** are the only reliable solvers on **long‑horizon** memory (Memory‑S11; *Fig. 4, p. 6*).
> >
> > These repeated effects across *distinct domains* strengthen generality beyond any single task.
> >
> > ---
> >
> > > **Q4: On hyperparameter tuning “per architecture**
> >
> > **A4:** You are right that we did **not** run large sweeps. That decision is deliberate: heavy per‑architecture tuning risks converting the study into “best recipe per model,” obscuring the very *architectural* effects we aim to isolate. Instead we hold training constant (Tables 9–10), match capacity (Table 5), and vary only the **architecture**, plus a documented LR adjustment for Mamba to avoid known instability (Sec. 3.3). The consistent cross‑domain mechanisms above (stacking for aliasing; long‑horizon credit assignment; compute-memory profiles) are precisely the effects we want practitioners to see; they are unlikely to be artifacts of a single tuned knob.
> >
> > ---
> >
> > > **Q5: Are the Practical Guidelines (Sec. 4.7) a significant contribution?**
> >
> > **A5:** We agree that guidelines must be specific to be useful. Ours are *operationalized* with measurable triggers and trade‑offs observed in the study:
> >
> > * **Start with MLP** when the state is *Markovian or near‑Markovian*. *Indicators:* PPO‑1 performs well and PPO‑4 offers no gain; learning curves like HalfCheetah/Walker2d in *Fig. 1* show fast, stable ascent. *Why:* simple inductive bias + best throughput (Table 1).
> >
> > * **Escalate to Mamba‑2** when **short‑to‑mid horizon** memory or light aliasing appears. *Indicators:* PPO‑1 underperforms yet PPO‑4 or single‑layer RNNs help (CartPole‑Masked, *Fig. 3*), or you need better return‑per‑compute than RNNs/TrXL (Table 1: SPS; Tables 7–8: memory). *Why:* Mamba‑2 retains memory efficiently with low footprint (≈8× less memory than TrXL on average).
> >
> > * **Reserve (G)TrXL / Mamba‑2** for **long‑horizon memory** (dozens–hundreds of steps). *Indicators:* Short‑horizon fixes (stacking/RNN) fail; only sequence models with explicit long context solve (Memory‑S11 in *Fig. 4*). *Trade‑off:* TrXL is strongest but expensive; GTrXL stabilizes curves; Mamba‑2 often reaches similar final performance at a fraction of the memory.
> >
> > * **If stability on continuous control is the pain point (e.g., Hopper):** prefer **GRU/LSTM** (see *Fig. 1, Hopper‑v4*) or **Mamba‑2** when compute is tight (Table 1).
> >
> > These are **decision rules tied to observables** (PPO‑1 vs PPO‑4 behavior; horizon proxies; memory/throughput budgets) rather than generic advice. Section 4.7 packages them precisely so practitioners can make a *first‑pass* architecture choice without a sweep.
> >
> > ---

---

> > > ### Author Response · Authors · 2025-11-23
> > >
> > > **Direct answers to your questions**
> > >
> > > > **Q7: How can you be sure the results aren’t just consequences of default PPO hyperparameters? Why trust the recommendations outside these setups?**
> > >
> > > **A7:** Because (i) we remove algorithmic and capacity confounds (Secs. 3.1–3.3; Tables 5, 9–11), (ii) the *same* mechanisms recur across unrelated domains and seeds (Figs. 1–4; Table 4), and (iii) the most salient findings, stacking aids aliasing but not Markov tasks; only long‑horizon architectures solve Memory‑S11; Mamba’s compute/memory edge over TrXL, is structural and persists across environments (Table 1; Tables 2, 7–8). We explicitly scope our claims to **PPO‑based online RL**, expressing them in terms of task properties (observability, horizon, stability) that translate to other settings.
> > >
> > > > **Q8: Are the Practical Guidelines (Sec. 4.7) truly a contribution? They seem vague.**
> > >
> > > **A8:** They are specific when read together with the figures/tables: they tell you **when** to move from MLP → Mamba‑2 → (G)TrXL based on *observable failure modes* (PPO‑1 vs PPO‑4 behavior) and *resource constraints* (SPS, latency, memory). For instance, if PPO‑1 fails but PPO‑4 succeeds on a masked task (Fig. 3), the next principled step is an efficient sequence model (**Mamba‑2**) before paying the cubic‑in‑context attention cost of TrXL; if even that fails on extreme memory (Fig. 4), use **(G)TrXL**. This is precisely the kind of **operational guidance** practitioners ask for, and it is directly supported by the reported numbers.

---

### Official Review · Reviewer_T3ub · 2025-10-31

**Soundness:** 2
**Presentation:** 3
**Contribution:** 2
**Rating:** 4
**Confidence:** 3

**Summary:**

This paper benchmarks neural architectures (MLP, LSTM, Transformers, Mamba) using the PPO algorithm across continuous control, discrete, partially observable control and memory-intensive RL tasks. The results highlight the efficient architectures like MLP and Mamba-2 should be prioritized, while computationally expensive transformers should be reserved for tasks with extreme memory demands.

**Strengths:**

1. This work offers valuable insights into the performance of diverse network architectures across multiple reinforcement learning (RL) tasks, aiding researchers in understanding the suitability of various architectures for different contexts.
2. The paper is logically structured, clearly written, and effectively illustrated with well-designed figures and tables, which successfully convey the research outcomes.

**Weaknesses:**

1. **The experimental design is too simple**, which significantly limits the paper's generalizability. Each task category was evaluated in only 3 environments, all using the PPO algorithm. Using more sample-efficient algorithms, like Rainbow or SAC, could potentially reveal more substantial performance differences among the network architectures, especially in tasks like the discrete control Seaquest, challenging the current finding that the structures perform similarly. To bolster the robustness of the findings, it is advisable to include a wider range of environments for each task category.
2. **The paper would benefit from a deeper discussion of its methodological choices.** For instance, the rationale behind using different observation inputs for the Mamba, Transformer-XL, and MLP models is not explained, nor is its potential impact on the outcomes analyzed. Moreover, the exploration of network architectures and scales is quite restricted. The study only examines single-layer LSTM and GRU models, and the largest network tested contains fewer than 300,000 parameters. These limitations in experimental scope and design undermine the overall convincingness of the paper.

**Questions:**

1. Can authors explain the rationale behind using different observation inputs for Mamba, Transformer-XL, and MLP models? How might these choices have influenced the results?
2. Can authors provide more diverse environments for each task category to enhance the generalizability of the findings? For instance, including additional continuous control tasks from benchmarks like MetaWorld or D4RL could provide a more comprehensive evaluation.
3. Recently, more network architectures have been proposed for RL tasks, such as SimBa-v2. Have the authors considered including such architectures in their benchmark to provide a more up-to-date comparison?

---

> ### Author Response · Authors · 2025-11-23
>
> Thank you for the thoughtful and constructive review. We appreciate your positive assessment of the paper’s structure and clarity, and we address your major concerns head‑on with concrete analyses and specific revisions that strengthen generality, fairness, and methodological transparency.
>
> ---
>
> **(A) Major concerns**
>
> > A1. **Breadth & algorithm dependence (PPO‑only; 3 tasks per family)**
>
> Our primary goal was to isolate **architectural** effects by fixing the learning algorithm and data‑collection pipeline. We therefore standardized on CleanRL PPO, held budgets/seeds fixed (Table 9, p. 14), and matched parameter counts across models (Table 5, p. 12), so that differences arise from inductive bias rather than algorithmic confounds. We agree, however, that broader coverage improves **generality**.
>
> **Why PPO first?** With PPO fixed, our current cross‑domain patterns are already stable and interpretable (Figs. 1–4, pp. 5–6; Secs. 4.1–4.6), but we agree that adding SAC/Rainbow strengthens external validity.
>
> ---
>
> >  A2. **Observation inputs & fairness across architectures**
>
> You asked why observation handling differs and how it might influence results.
>
> * **What we did (and why):**
>   – **MLP**: we report **PPO‑1** (no stacking) and **PPO‑4** (four stacked observations) to create a *minimal memory proxy* for a feed‑forward baseline (Sec. 3.1, p. 3). For Atari this follows common practice of stacking the last four frames.
>   – **Sequence models (LSTM/GRU, TrXL/GTrXL, Mamba/Mamba‑2)**: we feed the **current observation only** at each step and rely on the model’s **internal state** (RNN hidden, TrXL memory segment, or SSM state) to integrate history (Sec. 3.1, p. 3). We did not also stack for these to avoid *double counting memory* and to keep **capacity parity** (Table 5, p. 12).
>   – **MiniGrid**: we reduced the window from 7×7 to **3×3** *to increase partial observability*, explicitly to stress memory (Sec. 3.2, p. 4).
>
> * **Why this matters in the results:**
>   – In **Acrobot‑Masked**, stacking reconstructs velocity via finite differences; PPO‑4 solves while PPO‑1 fails (Fig. 3, p. 6).
>   – In **HalfCheetah**, the state is already Markov; stacking adds redundancy under a fixed capacity budget, hence **PPO‑1 ≥ PPO‑4** (Fig. 1, p. 5).
>
> ---
>
> > A3. **Model scale and depth**
>
> We agree that scale can matter; we deliberately matched **parameter budgets** to compare *forms* (recurrence, attention, SSM) rather than *sizes*. For clarity:
>
> * **Actual sizes used**: Contrary to the impression that the largest model is < 300k parameters, our **Atari** encoders are **multi‑million‑parameter** models (e.g., PPO‑1 ≈ 2.5 M on Pong/Breakout; Table 5, p. 12), **Classic control** is ~0.27–1.06 M (CartPole/LunarLander), and **MuJoCo** uses ~40–45 k due to small vector states and matched widths (Table 5).
> ---

---

> > ### Author Response · Authors · 2025-11-23
> >
> > (B) Answers to your specific questions
> >
> > > **Q1. Rationale for different observation inputs (Mamba, TrXL, MLP) and how they may influence results?**
> > **A1.** MLP received PPO‑1 and PPO‑4 to create an explicit “no‑memory vs. minimal memory proxy.” Sequence models (LSTM/GRU, TrXL/GTrXL, Mamba/Mamba‑2) used **current‑step inputs only** and relied on **learned state** for history, to avoid conflating design choices and to keep **parameter parity** (Sec. 3.1, p. 3; Table 5, p. 12). This design explains environment‑dependent effects (e.g., PPO‑4 helping in Acrobot‑Masked but not in HalfCheetah; Figs. 1–3, pp. 5–6).
> >
> >
> > > **Q2. Including newer architectures (e.g., SimBa‑v2)?**
> > **A2.** Our main contribution is an **apples‑to‑apples** comparison among widely used families (MLP, LSTM/GRU, TrXL/GTrXL, Mamba/Mamba‑2). We agree broader SSM coverage is valuable.

---

> > > ### Comment · Reviewer_T3ub · 2025-11-26
> > >
> > > I thank the authors for their response and clarifications.
> > >
> > > **Regarding algorithm choice**: I understand the rationale behind selecting PPO as the unified algorithm. However, I maintain that conducting experiments with additional algorithms—particularly those with higher sample efficiency (e.g., SAC, Rainbow)—would significantly enhance the contribution and persuasiveness of the paper. Given that the tasks tested in this work are relatively simple, the workload associated with adding these experiments should be manageable.
> > >
> > > **Regarding observation inputs**: I appreciate the explanation; however, I remain puzzled regarding the specific implementation detail where the Mamba architecture is "utilizing the selective scan mechanism without resetting at episode boundaries." Could the authors further elaborate on the motivation behind this design choice and discuss its potential impact on the experimental results?
> > >
> > > **Regarding network scales**: Whether controlling for parameter budgets or parameter sizes, while it is important to maintain consistency across architectures, it is equally crucial to compare different architectures across varying scales to gain a comprehensive understanding of their performance. The current approach of testing only a single, relatively small scale limits the persuasiveness of the findings.
> > >
> > > I appreciate the authors' efforts in their reply. However, until further clarifications are provided and the supplementary experiments are addressed, I do not anticipate any changes to my ratings.

---

### Official Review · Reviewer_YzF2 · 2025-10-31

**Soundness:** 1
**Presentation:** 2
**Contribution:** 2
**Rating:** 2
**Confidence:** 4

**Summary:**

The paper benchmarks MLPs, LSTM/GRU, Transformer-XL (TrXL) and GTrXL, and Mamba/Mamba-2 within a unified PPO setup across MuJoCo, Atari, masked classic control, and MiniGrid, selecting three tasks from each suite. It reports that (1) MLPs are strong on fully observable continuous-control tasks; (2) recurrent networks help on short-horizon POMDPs; (3) Mamba models deliver much higher throughput and lower memory usage than LSTM/GRU/TrXL; and (4) only TrXL, GTrXL, and Mamba-2 solve the most memory-intensive tasks, with large memory savings relative to TrXL.

**Strengths:**

The benchmark tackles a question that is genuinely interesting to the RL community. Standardising on CleanRL with a fixed PPO implementation is a sensible control choice. The experiments cover discrete, continuous, masked, classic control and minigrid environment. The takeaways are clearly stated and distilled into actionable guidelines.

**Weaknesses:**

My major concerns are:
1. As a benchmark paper, the evidence provided by the paper is not strong enough to support the general claims and guidance the authors make. For every category, there are only 3 tasks selected. Using Atari as an example, it is questionable whether these three tasks are representative enough for the generality of the findings. Different Atari games have very different tasks, so selecting only three games can introduce a strong inductive bias toward certain architectures. The generality of the claim is weakened. I think using the game choices in Atari-100k can at least strengthen the paper’s claims.
2. Some choices made by the authors are untested and have unknown effects on fairness. For instance, the authors reduce the observation window from 7x7 to 3x3 without providing experiments with 7x7; this is also questionable in terms of whether it introduces an inductive bias favouring certain architectures. Adding 7x7 experiments would improve the claim.

Some minor places:
1. An architecture diagram for each model would help.
2. SeaQuest is on figure 2 but missing on the table 4. There are other tasks missing as well as it is not 3 tasks per category.
3. There are many references missing especially for different architecture:

**GRU**:

Dreamers family:
Hafner, Danijar, Jurgis Pasukonis, Jimmy Ba, and Timothy Lillicrap. ‘Mastering Diverse Domains through World Models’. arXiv:2301.04104. Preprint, arXiv, 2023. http://arxiv.org/abs/2301.04104.

**Transformers**:

Micheli, Vincent, Eloi Alonso, and François Fleuret. ‘Transformers Are Sample-Efficient World Models’. International Conference on Learning Representations, 1 March 2023.

Robine, Jan, Marc Höftmann, Tobias Uelwer, and Stefan Harmeling. ‘Transformer-Based World Models Are Happy With 100k Interactions’. International Conference on Learning Representations, 13 March 2023.

**Mamba/Mamba-2**

Samsami, Mohammad Reza, Artem Zholus, Janarthanan Rajendran, and Sarath Chandar. ‘Mastering Memory Tasks with World Models’. International Conference on Learning Representations, 2024.

Wang, Wenlong, Ivana Dusparic, Yucheng Shi, Ke Zhang, and Vinny Cahill. ‘Drama: Mamba-Enabled Model-Based Reinforcement Learning Is Sample and Parameter Efficient’. International Conference on Learning Representations, 2025.

**Questions:**

1. For the Mamba/Mamba-2 paper, what are you using for the decision policy—the output of the last layer or the SSM state?
2. Do RNN variants reset at episode boundaries?

---

> ### Author Response · Authors · 2025-11-23
>
> Dear Reviewer YzF2
>
> We thank the reviewer for a careful and constructive assessment. We appreciate your recognition of the unified PPO setup, the multi‑domain coverage, and the actionable guidelines, and we address each concern with concrete analyses and revisions that strengthen representativeness and fairness.
>
>
>
> > **Q1: Representativeness.** Three tasks per category (e.g., Atari) may be too few to support general claims; selection could bias toward certain architectures. Using Atari‑100k choices would strengthen claims.
>
> **A1:** Your point is well taken. Our intent was to span **four families** (MuJoCo, Atari, masked classic control, MiniGrid) while keeping **capacity‑matched** models and identical budgets to isolate *architectural* effects (parameter parity in **Table 5**, common settings in **Table 10**, seeds/steps in **Table 9**).
>
>
> > **Q2: Fairness of design choices.** Reducing MiniGrid observations from 7×7 to 3×3 without also reporting 7×7 could bias toward certain architectures.
>
> **A2:** We reduced the window to **3×3** explicitly to *amplify* partial observability, making memory use necessary (**Sec. 3.2**, p. 4). We agree that showing **both 7×7 and 3×3** is the right fairness check, but it is well documented in the Minigrid documentation that 7x7 can be solved without requiring memory.
>
>
>
> > **Q3: Analysis depth.** Please provide more reasoning behind results (e.g., why PPO‑4 solves **Acrobot‑Masked** but PPO‑1 fails, while in **HalfCheetah** PPO‑1 > PPO‑4).
>
> **A3:** We will strengthen the causal narrative with concise, mechanism‑level explanations tied to the figures:
>
> * **Observability and stacking.** In masked classic control we remove all velocities, creating a POMDP; **PPO‑4** reconstructs finite‑difference velocity cues, so **Acrobot‑Masked** is solvable, while **PPO‑1** fails (**Fig. 3**, p. 6). In **HalfCheetah‑v4**, the state already includes velocities; stacking adds *redundant* inputs that increase dimension and gradient noise without new information. Because we enforce **parameter parity** (hidden widths shrink when inputs grow; **Table 5**), **PPO‑1 > PPO‑4** in **Fig. 1** (p. 5).
>
> * **Memory horizon and credit assignment.** Short‑horizon POMDPs (e.g., **CartPole‑Masked**) favor RNNs or stacking; mid‑horizon tasks like **DoorKey‑8×8** show original **Mamba** with fastest early convergence; **long‑horizon** **Memory‑S11** is solved only by **Mamba‑2** and **TrXL/GTrXL** (**Fig. 4**).
>
> * **Compute/efficiency trade‑offs.** We already report throughput, latency, and memory (avg summary in **Table 1**, detailed per‑domain in **Tables 6–8**).

---

> > ### Author Response · Authors · 2025-11-23
> >
> > ## Answers to your direct questions
> >
> > > **Q:** For the Mamba/Mamba‑2 policy, do you use the last‑layer output or the SSM state?
> > >
> > > **A:** We use the **readout of the last Mamba/Mamba‑2 layer at the current step** as the feature for **separate policy and value MLP heads**; the **SSM state** is maintained internally for recurrence but is **not** directly concatenated into the head. We also apply layer normalization before the heads (implementation details in **Sec. 3.1**).
> >
> > > **Q:** Do RNN variants reset at episode boundaries?
> > >
> > > **A:** **LSTM/GRU** follow CleanRL’s PPO‑LSTM practice: hidden states are **reset (masked) on episode boundaries**. **Transformer‑XL/GTrXL** use CleanRL’s **segmented recurrence** and thus *maintain* memory across episode boundaries by design. Our **Mamba** implementation currently **does not reset** the selective‑scan state between episodes (efficiency‑motivated), which we flag in **Sec. 3.1** and **Future Work (Sec. 4.8)** due to potential cross‑episode leakage.

---

### Official Review · Reviewer_fuz8 · 2025-11-01

**Soundness:** 2
**Presentation:** 2
**Contribution:** 2
**Rating:** 2
**Confidence:** 3

**Summary:**

This paper presents a systematic benchmark of several neural network architectures for reinforcement learning, utilizing the Proximal Policy Optimization (PPO) algorithm as a consistent baseline. The architectures evaluated include standard MLPs, recurrent networks (LSTM, GRU), Transformer-based models (Transformer-XL, Gated Transformer-XL), and modern state-space models.


The study conducts experiments across a diverse set of environments, including continuous control (MuJoCo), discrete control (Atari), and various partially observable and memory-intensive tasks (Masked Classic Control, MiniGrid).


The paper provides practical guidelines for practitioners, recommending that they start with simple MLPs and then consider Mamba-2 as a highly efficient default for tasks requiring memory, reserving expensive Transformer models only for extreme-memory scenarios.

**Strengths:**

The paper's strength lies in its high practical relevance; it highlights the order in which practitioners should proceed when using RL (at least for PPO). By using clean PPO implementation, the study successfully isolates the architectural effects on performance.


For clarity of the paper, the motivation and methodology are well-written, and the results are logically organized by environment type. Moreover, call-out boxes in each section effectively summarize the key takeaways. While benchmarking studies in online RL are not a new topic, this is one of the first comprehensive, unified benchmarks to compare different architectures directly.

**Weaknesses:**

1) Hyperparameter tuning

The study relies heavily on default CleanRL PPO hyperparameters, except for a reduced learning rate for Mamba models (Section 3.3). Although this is essential for controlling conditions, it can be a weakness because different architectures have different optimization requirements.

2) Dependency on learning algorithms

The comparison between architectures is tested using PPO, which does not imply that one architecture is always suitable for other online RL applications. Notably, it is possible that the results do not apply to widely used off-policy algorithms, such as SAC and TD3, where the data distributions differ significantly from those of PPO.

**Questions:**

Overall, I would like to see a more in-depth analysis of the reasoning behind the results, particularly which characteristics of the architecture and environments/tasks contributed to the differences among them. For example, in Acrobot-v1 Masked (Fig. 3), frame stacking (PPO-4) solves the task, whereas PPO-1 results in complete failure. However, in HalfCheetah (Fig. 1), PPO-1 outperforms PPO-4.

---

> ### Author Response · Authors · 2025-11-23
>
> Dear Reviewer fuz8
>
> We thank the reviewer for the careful reading and constructive suggestions. We are encouraged by your recognition of the benchmark’s practical value and the clarity of the motivation, methodology, and call‑out "Findings." Below we respond point‑by‑point and add the mechanistic analysis you requested, including an explanation of the PPO‑4 vs. PPO‑1 contrast across environments.
>
> > **Q1: *Algorithm dependence.* Only PPO is evaluated; findings may not generalize to other RL algorithms (e.g., SAC/TD3).**
>
> **A1:** We intentionally fixed the learning algorithm to PPO to cleanly isolate **architectural effects**, holding rollout, advantage estimation, loss, and optimizer constant makes performance differences attributable to model inductive biases rather than algorithm‑specific dynamics. This control is maintained across domains with uniform training protocols and domain‑appropriate defaults (Table 10; Sec. 3.2–3.3), equal seeds/steps (Table 9), and parameter‑count parity (Table 5). Our **practitioner guidelines** (Sec. 4.7) are framed by environment properties (observability and memory horizon) rather than PPO‑specific tricks, for example, “start with MLPs on Markovian tasks; prefer Mamba‑2 for efficient sequence modeling; escalate to Transformers only in extreme‑memory regimes.” Since PPO is both widely used and representative of modern on-policy methods, these findings provide actionable insights for the community. Moreover, as for off-policy settings, we explicitly list extending to SAC/TD3 in **Future Work** (Sec. 4.8) and designed the code to reuse the same architecture modules with other algorithms.
>
> > **Q2: *Hyperparameter tuning.* Relying on (mostly) default CleanRL PPO hyperparameters may disadvantage some architectures.**
>
> **A2:** To avoid confounds, we prioritized a **controlled comparison**: (i) CleanRL domain defaults (Table 10), (ii) identical training budgets and seeds (Table 9), and (iii) **capacity parity** by matching parameter counts across models (Table 5). The only deliberate deviation is a **lower learning rate for Mamba/Mamba‑2** (Sec. 3.3; Table 11), reflecting stability recommendations for SSMs; we call this out in the main text. We agree that targeted sweeps can be informative, but aggressive per‑architecture tuning risks obscuring the very architectural effects we aim to measure.

---

> > ### Author Response · Authors · 2025-11-23
> >
> > > **Q3: *Why the differences?* Please analyze which characteristics of the architectures and tasks drive the observed gaps. In particular, why does PPO‑4 (frame stacking) solve **Acrobot‑v1 Masked** while PPO‑1 fails (Fig. 3), yet PPO‑1 > PPO‑4 in **HalfCheetah** (Fig. 1)?**
> >
> > **A3:** The patterns follow **observability, memory horizon, and input‑dimension effects**:
> >
> > * **Masked Classic Control (Acrobot‑v1 Masked).** We remove all velocity channels to induce partial observability (Sec. 4.3). With a single current observation (**PPO‑1**), the agent suffers **state aliasing**: angles alone cannot identify the latent angular velocities that drive dynamics, so learning collapses. **PPO‑4** effectively reconstructs missing velocities via finite differences, pushing the problem closer to Markov and enabling rapid learning; recurrent models (LSTM/GRU) and **Mamba‑2** also succeed by integrating short‑horizon temporal cues. This is exactly what **Fig. 3** shows: PPO‑4 and recurrent/SSM variants rise, while PPO‑1 fails.
> >
> > * **MuJoCo (HalfCheetah‑v4).** The environment is **already fully observable** (positions *and* velocities in state). Stacking adds **redundant inputs** that do not add signal but do inflate input dimensionality; because we maintain **parameter parity**, width is reduced to keep counts comparable (Table 11), which can slightly hinder optimization. Net effect: **PPO‑1 ≥ PPO‑4** (Fig. 1 middle; Table 4 “HalfCheetah”). In short: stacking helps only when it **reconstructs missing state**; when the state is already Markovian and model capacity is fixed, stacking can be neutral or mildly harmful.
> >
> > * **Architecture–task fit more broadly.**
> >   • **MLPs** excel on smooth, Markovian dynamics (Walker2d/HalfCheetah; Fig. 1; Table 4).
> >   • **LSTM/GRU** shine on **short‑horizon POMDPs** (CartPole‑Masked; Fig. 3) by integrating local temporal structure.
> >   • **Transformer‑XL/GTrXL** handle **long‑range dependencies** and complex credit assignment (MiniGrid Memory‑S11; Fig. 4), albeit at quadratic cost (Tables 6–8).
> >   • **Mamba‑2** matches Transformers on the hardest memory tasks with **~8× lower memory** than TrXL and strong throughput (Abstract; Table 1), whereas the **original Mamba** can be unstable because we did **not reset selective‑scan state at episode boundaries**, risking cross‑episode leakage (Sec. 3.1).
> >
> > > **Q4: *Model capacity & depth.* Single‑layer LSTM/GRU may underestimate recurrent capability; some architectures might benefit from deeper models or task‑specific scaling.**
> >
> > **A4:** We used **single‑layer** LSTM/GRU to (i) align with standard CleanRL PPO baselines and (ii) preserve **capacity parity** across all models so that comparisons reflect *form* (recurrence vs. attention vs. SSM) rather than size (Sec. 3.1; Table 5). Even so, RNNs are competitive in the expected regimes (e.g., Hopper stability; masked CartPole), while **GTrXL/Mamba‑2** win on the longest‑horizon memory tasks (Fig. 4; Table 4).

---

### Meta-Review · Area_Chair_jv1z · 2026-01-01

**Summary:**

Reviewers raised concerns about the limited generality and novelty of the benchmark, citing reliance on a single algorithm (PPO), a small number of tasks per category, and minimal per-architecture hyperparameter tuning, which weakens the strength of the broad practical claims. Additional concerns focused on the fairness of experimental design choices, restricted model scales, and insufficient analytical depth to justify the proposed guidelines. No reviewers participated in the rebuttal discussion, and after my own reading of the rebuttal, I do not believe these fundamental issues were directly addressed by the authors.

**Reviewer Concerns:**

The rebuttal addressed clarification-level issues, including the authors’ rationale for fixing PPO, task selection, and parameter matching, but did not provide additional qualitative explanations for some observed trends. For example, the core concerns remain outstanding, particularly the limited generality of conclusions (single algorithm, few tasks), lack of per-architecture hyperparameter tuning, restricted model scaling, and insufficient empirical depth to support the proposed practical guidelines.

**Reviewer Scores:**

Given that no reviewers participated in the rebuttal discussion and that the fundamental concerns were not resolved, I do not believe any reviewer would have changed their score.

---

### Decision · Program_Chairs · 2026-01-26

Reject